# GnRH Induces Citrullination of the Cytoskeleton in Murine Gonadotrope Cells

**DOI:** 10.3390/ijms25063181

**Published:** 2024-03-10

**Authors:** Elizabeth B. Quigley, Stanley B. DeVore, Shaihla A. Khan, Zachary M. Geisterfer, Heather M. Rothfuss, Ari O. Sequoia, Paul R. Thompson, Jesse C. Gatlin, Brian D. Cherrington, Amy M. Navratil

**Affiliations:** 1Department of Zoology and Physiology, University of Wyoming, Laramie, WY 82071, USA; equigle2@uwyo.edu (E.B.Q.); atourtel@uwyo.edu (A.O.S.); anavrati@uwyo.edu (A.M.N.); 2Department of Pediatrics, University of Cincinnati College of Medicine, Division of Asthma Research, Cincinnati Children’s Hospital Medical Center, Cincinnati, OH 45229, USA; devoresb@mail.uc.edu; 3Genus PLC, DeForest, WI 53532, USA; shaihla.khan@gmail.com; 4Department of Cell Biology, Duke University School of Medicine, Durham, NC 27710, USA; zmg2@duke.edu; 5Department of Biochemistry and Molecular Pharmacology, University of Massachusetts Medical School, Worcester, MA 01605, USA; Paul.Thompson@umassmed.edu; 6Department of Molecular Biology, University of Wyoming, Laramie, WY 82071, USA; jgatlin@uwyo.edu

**Keywords:** gonadotrope, peptidylarginine deiminase, citrullination, cytoskeleton, β-tubulin, β-actin, microtubules

## Abstract

Peptidylarginine deiminases (PADs or PADIs) catalyze the conversion of positively charged arginine to neutral citrulline, which alters target protein structure and function. Our previous work established that gonadotropin-releasing hormone agonist (GnRHa) stimulates PAD2-catalyzed histone citrullination to epigenetically regulate gonadotropin gene expression in the gonadotrope-derived LβT2 cell line. However, PADs are also found in the cytoplasm. Given this, we used mass spectrometry (MS) to identify additional non-histone proteins that are citrullinated following GnRHa stimulation and characterized the temporal dynamics of this modification. Our results show that actin and tubulin are citrullinated, which led us to hypothesize that GnRHa might induce their citrullination to modulate cytoskeletal dynamics and architecture. The data show that 10 nM GnRHa induces the citrullination of β-actin, with elevated levels occurring at 10 min. The level of β-actin citrullination is reduced in the presence of the pan-PAD inhibitor biphenyl-benzimidazole-Cl-amidine (BB-ClA), which also prevents GnRHa-induced actin reorganization in dispersed murine gonadotrope cells. GnRHa induces the citrullination of β-tubulin, with elevated levels occurring at 30 min, and this response is attenuated in the presence of PAD inhibition. To examine the functional consequence of β-tubulin citrullination, we utilized fluorescently tagged end binding protein 1 (EB1-GFP) to track the growing plus end of microtubules (MT) in real time in transfected LβT2 cells. Time-lapse confocal microscopy of EB1-GFP reveals that the MT average lifetime increases following 30 min of GnRHa treatment, but this increase is attenuated by PAD inhibition. Taken together, our data suggest that GnRHa-induced citrullination alters actin reorganization and MT lifetime in gonadotrope cells.

## 1. Introduction

Arginine residues in proteins can be converted to the non-coded amino acid citrulline through a post-translational modification (PTM) called deimination or citrullination. This reaction is catalyzed by a family of enzymes termed peptidylarginine deiminases (PADs or PADIs), which are composed of five isoforms: PAD1-4 and 6 [1,2,3]. Each PAD family member appears to have preferential catalytic targets for citrullination with the exception of PAD6, which lacks enzymatic activity [4,5]. PADs are expressed in a wide array of normal tissues such as the brain, skin, and female reproductive tissues, suggesting that citrullination also has an important function in regulating physiological processes [1,3,6,7,8].

Anterior pituitary gonadotrope cells are a critical component of the hypothalamic–pituitary–gonadal axis. Gonadotropin-releasing hormone (GnRH) is secreted in a pulsatile fashion from hypothalamic neurons and binds to GnRH receptors located on the plasma membrane of gonadotropes. Our previous work shows that PAD2 is highly expressed relative to other PAD isoforms in the gonadotrope-derived LβT2 cell line, as well as female murine gonadotrope cells, and is significantly elevated during the estrus phase of the estrous cycle when ovulation is occurring [9]. These initial studies raised additional questions as PAD2 is also localized to the cytoplasmic compartment of the cell. A logical consequence of this localization is that PAD2 also targets cytoplasmic proteins for citrullination that may contribute to gonadotrope function.

Cytoplasmic cytoskeletal proteins such as actin, tubulin, vimentin, and keratins are targets of PADs; however, little is currently known about the effects of citrullination on cytoskeletal function [10,11,12]. The citrullination of keratins (K1 and K10) promotes differentiation of the human epidermis [13], while vimentin citrullination results in increased solubility and filament disassembly [14]. Citrullinated proteins, including those that compose cytoskeletal filaments, can trigger the production of autoantibodies called anti-citrullinated protein antibodies (ACPAs), which are common in many autoimmune disorders including rheumatoid arthritis [15,16]. Currently, the physiological versus pathological consequences of citrullination on cytoskeletal function are unknown in most cell types including gonadotropes.

Rapidly after GnRH binding, the actin cytoskeleton in the gonadotrope dramatically reorganizes, forming lamellipodia and filopodia. In primary pituitaries, engagement of the actin cytoskeleton is necessary for the spatial positioning of gonadotropes towards vasculature and LH release [17,18]. Following hormone release, new secretory vesicles must then be trafficked along microtubules (MTs) to await successful docking with the plasma membrane. Recent evidence suggests that MTs are targets of extensive PTMs, which can stabilize MTs and may dramatically alter the MT landscape, thereby providing navigational cues for specific molecular motors [19,20,21]. What remains unclear is how PTMs, like citrullination, are utilized by gonadotropes to alter cytoskeletal function and dynamics.

Utilizing mass spectrometry (MS), we identified a GnRH agonist (GnRHa)-induced temporal citrullinome in the gonadotrope-derived LβT2 cell line. Of our identified proteins, we specifically examined actin and tubulin since previous data show that they are important for gonadotrope function and hormone secretion [22,23,24,25]. Our results show that the GnRHa treatment of LβT2 cells stimulates the citrullination of actin and tubulin in distinct temporal patterns and that citrullination of both filaments is blocked by the PAD inhibitor biphenyl-benzimidazole-Cl-amidine (BB-ClA). Imaging analysis of LβT2 cells and dispersed mouse gonadotropes demonstrates that inhibiting citrullination disrupts the reorganization of β-actin following GnRHa stimulation. Using fluorescently tagged end binding protein 1 (EB1-GFP), we show that MT polymerization speed is not altered by citrullination; however, MT average lifetimes increase following 30 min of GnRHa treatment compared to the control but are blunted following PAD inhibition. Collectively, our results are the first to show that GnRHa rapidly induces PAD catalyzed citrullination to regulate cytoskeletal architecture and dynamics in gonadotropes.

## 2. Results

### 2.1. GnRHa Stimulation of LβT2 Cells Rapidly Induces Protein Citrullination

Our previous work discovered that GnRHa stimulation of LβT2 cells results in the citrullination of histone H3, which alters gonadotropin gene expression [9]. To identify additional citrullinated proteins beyond histones, LβT2 cells were serum starved and then treated with 10 nM GnRHa for 0, 10, and 30 min. Equal concentrations of LβT2 lysates were labelled with vehicle as a background control or a citrulline specific biotin-phenylglyoxal probe (PG) to enrich citrullinated proteins as previously described [26,27]. The samples were examined using MS and analyzed, with the treatment groups normalized to the controls. Our MS data revealed that citrullinated proteins are temporally associated with distinct cellular pathways following 10 and 30 min of GnRHa stimulation. After 10 min of GnRHa treatment, there is enrichment of citrullinated proteins in pathways involved in cellular proliferation, immunoglobulin binding activity, transcriptional activation, and ubiquitin conjugation, (Figure 1A). After 30 min of GnRHa treatment, enrichment of citrullinated proteins is seen in pathways involved in protein trafficking to the endoplasmic reticulum, mitotic cell cycle progression, RAS/RAF/MEK/ERK signaling, and centrosome formation (Figure 1B). These results show that not only does GnRHa stimulation of LβT2 cells result in temporal protein citrullination but also citrullination is an important PTM involved in cellular pathways essential to gonadotrope function [28,29,30]. Interestingly, a number of these pathways are involved in the regulation of the cytoskeleton, including the activation and coordination of actin remodeling and MT stability [22,23,24,31,32,33].

### 2.2. Cytoskeletal Proteins Are Temporally Citrullinated following GnRHa Stimulation of LβT2 Cells

Our MS data reveals citrullination of cytoplasmic actin is elevated after 10 min of GnRHa treatment (Figure 2A). In addition, actin-associated proteins are also citrullinated and involved in processes such as actin nucleation and lamellipodia formation (Figure 2B). Interestingly, our MS analysis shows that citrullinated β-tubulin is elevated following 30 min of GnRHa stimulation (Figure 2C). Like actin, additional citrullinated proteins are involved in MT polymerization, organization, and intracellular transport (Figure 2D). Taken together, citrullination may be an important PTM in the regulation of cytoskeletal remodeling and dynamics.

### 2.3. GnRHa Stimulates the Citrullination of β-Actin to Alter Gonadotrope Cytoskeletal Architecture

Our work has previously shown that within minutes, GnRHa treatment of LβT2 cells and ex vivo murine pituitary slices results in dynamic actin cytoskeletal reorganization [22,34]. Despite these findings, the cohort of PTMs that GnRHa induces to initiate actin reorganization is unclear. Given that β-actin was identified in our MS studies, we validated this finding by first serum starving LβT2 cells and then treating them with 10 nM GnRHa for 0, 10, and 30 min. Citrullinated proteins were then enriched with a biotin-PG probe. The samples were examined using Western blot, and membranes were probed with an anti-β-actin antibody. Our results support that GnRHa stimulation of LβT2 cells rapidly induces β-actin citrullination, with increased levels occurring at 10 min. Quantification of multiple Western blots (*n* = 4) shows that 10 min of GnRHa stimulation results in a 2.5-fold increase in the citrullination of β-actin (Figure 3A). To confirm that the citrullination of β-actin is PAD dependent, LβT2 cells were pretreated with DMSO or 1 μM of PAD inhibitor BB-ClA for 12 h and then treated with 10 nM GnRHa for 0 or 10 min. Pre-treatment of LβT2 cells with BB-ClA resulted in a decrease in citrullinated β-actin as compared to the vehicle-treated controls (Figure 3B). These results establish that GnRHa rapidly induces PAD-catalyzed citrullination of β-actin in LβT2 cells, consistent with our MS results (Figure 2A).

We next examined the effect of β-actin citrullination on gonadotrope cytoskeletal architecture. LβT2 cells were serum starved and then treated with vehicle or 1 µM BB-ClA, followed by 10 nM GnRHa for 0, 10, and 30 min. Cells were fixed, probed with an anti-β-actin antibody and an appropriate secondary antibody, and DAPI, and then imaged using immunofluorescent confocal microscopy. Our results show that GnRHa-induced actin reorganization is blunted following pre-treatment with BB-ClA (Appendix A). Next, female murine pituitaries were explanted, dispersed, and subjected to the same treatment protocol described above and then imaged using an anti-LHβ antibody, followed by an appropriate secondary antibody (red) to identify gonadotropes. As we have previously shown, GnRHa stimulation results in reorganization of actin at 10 min with distinct lamellipodia that contain LH at the leading edge (Figure 4) [22]. Lamellipodia are not present in gonadotropes pre-treated with BB-ClA. These results are the first to show that PAD-catalyzed citrullination of actin participates in the GnRHa induced-cytoskeletal reorganization in gonadotropes.

### 2.4. GnRHa-Induced Citrullination of β-Tubulin Increases MT Lifetimes but Has No Effect on Polymerization Speed or Nucleation Rate

In addition to β-actin, our MS data revealed that β-tubulin is citrullinated following 30 min of GnRHa treatment. To validate this, LβT2 cells were serum starved and treated with 10 nM GnRHa for 0, 10, and 30 min, and then, citrullinated proteins were enriched with biotin-PG. Citrullinated protein samples and 5% input for each treatment were examined via Western blot and membranes probed with an anti-β-tubulin antibody. Western blots reveal that 10 nM GnRHa stimulation of LβT2 cells induces β-tubulin citrullination, with elevated levels occurring at 30 min (Figure 5A). Quantification of Western blots found that 30 min of GnRHa stimulation results in a two-fold increase in the citrullination of β-tubulin. Pre-treatment of LβT2 cells with 1 μM BB-ClA results in a decrease in the level of citrullinated β-tubulin following GnRHa stimulation (Figure 5B). Interestingly, these results show that 30 min of GnRHa induces the PAD-catalyzed citrullination of β-tubulin in LβT2 cells, which is a slower kinetic than that of β-actin.

While our data clearly show that GnRHa induces the citrullination of β-tubulin, how the citrullination of tubulin effects MT dynamics is currently unknown. Our initial imaging analysis of LβT2 and dispersed murine pituitary cells did not reveal gross changes in MT architecture with PAD inhibition. To address this limitation, we utilized a green fluorescent protein tagged end binding protein 1 (EB1-GFP) to visualize the growing plus end of MT dynamics in real time as previously described [35,36]. Thus, we transfected LβT2 cells for 48 h with EB1-GFP. Following transfection, cells were pre-treated for 12 h with DMSO or 1 µM BB-ClA, serum starved, and then received either vehicle or 10 nM GnRHa for 30 min. Transfected LβT2 cells were imaged at 0.5 s intervals for 65–94 s on an Olympus IX81 spinning disk confocal microscope (SDCM) using a 60X oil immersion objective (Appendix A). The EB1-GFP image series were analyzed in Fiji software version 2.14.0/1.54f and used to quantify MT nucleation rates. Kymograph analysis revealed that there is no significant change in the flux of comets per second between treatment groups, suggesting that citrullination of tubulin does not increase the nucleation of additional MTs (Figure 6).

Lastly, we examined whether the citrullination of tubulin effects MT polymerization rates and lifetimes. LβT2 cells were transfected as above, pre-treated for 12 h with DMSO or 1µM BB-ClA, serum starved, and then received either vehicle or 10 nM GnRHa for 30 min. LβT2 cells were imaged at 0.5 s intervals for 65–94 s on an Olympus SDCM using a 60X oil immersion objective. Polymerization rate and lifetimes were analyzed in Fiji software version 2.14.0/1.54f using the *kymographs* function of the kymograph plugin generated by J. Rietdorf and A. Seitz as previously described [37,38,39]. The images of the growing plus ends of MT comets were plotted over time to generate individual kymographs as shown in the top panel (Figure 7A). By using the *read velocities from tsp* function of the kymograph plugin described above, we were able to extract the lengths and lifetimes of each individual track, reported as dx sum and dy sum. These values were normalized for pixel size and frame rate capture to provide accurate distances (lengths) and total times of comets (lifetimes). Polymerization speeds of MTs were analyzed by dividing MT lengths by lifetimes. Our results show that the MT polymerization speed was not altered by citrullination (Figure 7B). Interestingly, GnRHa treatment increases the average MT lifetime, but this effect is reduced following PAD inhibition (Figure 7C). These findings suggest that the GnRHa-induced citrullination of tubulin may promote MT stability in gonadotrope cells.

## 3. Discussion

Our studies provide novel insights into the function of the PAD-catalyzed citrullination of cytoskeletal filaments in gonadotrope cells. Many female reproductive tissues express PAD enzymes and have important secretory functions that require precise coordination of the cytoskeleton [1,3,6,7,8,22,25]. For example, Akiyama et al. found that PAD2 colocalizes with prolactin-containing secretory vesicles in lactotrope cells [27,40]. PADs were also highly expressed in uterine tissue as compared to the other murine tissues examined, where they localized to uterine luminal and glandular epithelial cells [7,41]. Interestingly, these uterine epithelial cells secrete a wide range of substances, termed the histotroph, that are critical for the survival and development of the conceptus [42].

Currently, the functional significance of the citrullination of β-actin and β-tubulin is not well defined. In prostate cancer cells, the citrullination of actin is important for increased cellular microvesicle (MV) release. In this model, inhibition of PAD activity with BB-ClA attenuates citrullination of the actin cytoskeleton, reduces membrane reorganization, and inhibits MV release [11]. The functional consequences of β-tubulin citrullination is even less clear. Citrullinated β-tubulin is present in multiple tumor-derived cell lines, yet the significance of this is not known [10]. There is evidence that citrullinated actin and tubulin are associated with rheumatoid arthritis, where the citrullinated forms of these two proteins stimulate autoantibody production [43,44]. Actin and tubulin, in addition to numerous other proteins, are highly citrullinated in brains from Alzheimer’s disease patients as compared to age-matched healthy controls [45].

Our results are consistent with previous work identifying citrullinated cytoskeletal filaments in various biological systems [10,27,46,47]; however, very few studies have examined the effects of citrullination on cytoskeletal dynamics. Work by Inagaki et al. found that the calcium-dependent citrullination of the intermediate filament vimentin at the amino terminal head results in filament disassembly. These findings are hard to interpret in the context of normal cell physiology since citrullination was induced during apoptosis [14]. It is also important to point out that it is presently unclear as to whether the citrullination PTM is removed from functioning proteins or if it decreases as a result of protein turnover. To date, a de-citrullinase enzyme has not been identified but is of great interest in the PAD biology field. Although a growing body of work indicates that the citrullination of cytoskeletal filaments is important in physiologic and pathologic states, extensive work remains to develop a complete understanding of the role of cytoskeletal citrullination in cell physiology.

We found that GnRHa induces the citrullination of β-actin to alter the actin cytoskeletal architecture, while the citrullination of β-tubulin appears to increase MT lifetimes. Taken together, it is tempting to speculate that our data suggest that the GnRH-induced citrullination of the gonadotrope cytoskeleton may contribute to hormone vesicle trafficking and secretion. Our previous studies show that PADs citrullinate the histone H3 tail on arginine residues 2, 8, and 17 to regulate the expression of the *Lhb*, and PAD inhibition decreases LHβ synthesis [9]; thus, we cannot rule out the possibility that any decrease in LH secretion following GnRH stimulation is not due to both mechanisms. It is hypothesized that following GnRH binding to its receptor, the cytoskeleton reorganizes to allow gonadotropin-containing vesicles to fuse with the plasma membrane, thereby secreting the hormone into the extracellular space. Stimulation of gonadotropes with GnRH rapidly initiates the release of LH, and secretion is inhibited by cytochalasin B but not colchicine or vinblastine [24]. This result indicates that the inhibition of actin reorganization alters acute GnRH-induced LH secretion, but inhibition of MTs does not. These findings, interpreted in the context of our work, suggest that the citrullination of β-actin may facilitate acute LH release following GnRHa stimulation.

The studies presented here support the hypothesis that the citrullination of β-tubulin increases the lifetime of MTs, which may allow hormone-containing vesicles to move along stable filaments to replenish those vesicles that successfully docked with the plasma membrane [17,18]. We cannot rule out that citrullination may also provide routing signals to effectively transport new/reserve secretory vesicles to the membrane following a pulse of GnRH. Such an idea is not without precedent as the acetylation of MTs is critical for cytoskeletal dynamics and vesicle trafficking [48,49,50,51]. Clearly, work remains to understand the full role of cytoskeletal citrullination in cellular function in gonadotropes.

Our work advances our knowledge about PAD enzyme function in cytoskeletal dynamics in female reproductive cells. We propose that GnRH rapidly induces the PAD2-catalyzed citrullination of β-actin and β-tubulin to allow for dynamic cytoskeletal changes that are potentially necessary for hormone secretion. In conclusion, the citrullination of β-actin and β-tubulin is a novel PTM that may regulate gonadotrope cytoskeletal architecture and dynamics. Our findings also add important insight into the functional role of PAD inhibitors as therapeutics. Specifically, these molecules may block cytoskeletal reorganization necessary for cell division and metastatic progression. Overall, this work furthers our understanding of the diverse roles of PAD-catalyzed citrullination in health and disease.

## 4. Materials and Methods

### 4.1. Materials

The anti-PAD2 antibody was purchased from ProteinTech (12110-1-AP, Rosemont, IL, USA) while the anti-β-actin antibody was obtained from ThermoFisher Scientific (MA1-140, Waltham, MA, USA). The anti-β-tubulin antibody and GnRH agonist (GnRHa) Buserelin were purchased from Sigma-Aldrich (T8328, 68630-75-1, St. Louis, MO, USA). All fluorescently labeled Alexa Fluor secondary antibodies were purchased from Molecular Probes (Eugene, OR, USA), and horseradish peroxidase (HRP) secondary antibodies were purchased from Cell Signaling Technology (Danvers, MA, USA). The anti-LHβ antibody was obtained from the A.F. Parlow National Hormone and Peptide Program (Cat# AFP344191, RRID:AB_2784507). Matrigel was purchased from BD Biosciences (San Jose, CA, USA), and casein was purchased from Vector Laboratories Inc. (Newark, CA, USA). The 35 mm glass bottom culture dishes were purchased from Mat-Tek Corporation (Ashland, MA, USA). The PAD inhibitor, biphenyl-benzimidazole-Cl-amidine (BB-ClA), and biotin-phenylglyoxal (biotin-PG) molecule were a generous gift from Paul Thompson. TransIT-2020 Transfection Reagent (MIR 5400) was purchased from Mirus Bio (Madison, WI, USA). The Human EB1 GFP vector (JB131) was a gift from Tim Mitchison and Jennifer Tirnauer (Addgene plasmid # 39299; http://n2t.net/addgene:39299 (accessed on 29 March 2021); RRID: Addgene_39299).

### 4.2. Cell Culture

LβT2 cells, a generous gift from Dr. Pamela Mellon (University of California, San Diego, CA, USA), were maintained in high glucose DMEM containing 2 mM glutamine, 100 U penicillin/mL, 100 µg streptomycin/mL, and 10% fetal bovine serum (FBS) (HyClone, Logan, UT, USA). Phenol red-free media (SH30272.01) was purchased from HyClone. All cells were grown in 5% CO_2_ at 37 °C in a humidified environment.

### 4.3. Citrullinated Protein Purification Using Biotin-Phenylglyoxal (PG) Immunoprecipitation

LβT2 cells were incubated with phenol red-free media (DMEM/F-12 1:1 phenol red-free) for 6 h followed by treatment with vehicle or GnRHa (10 nM) for the indicated times. For the BB-ClA experiments, the LβT2 cells were pre-treated with DMSO or 1 µM BB-ClA for 12 h in phenol red-free media containing 2.5% charcoal-stripped FBS and then treated with 10 nM GnRHa for 0, 10 and 30 min. Following treatment, cells were washed from the plate in cold PBS then resuspended in HEPES buffer. Lysates were labelled with biotin-PG [27]. For biotin-PG immunoprecipitation, magnetic streptavidin agarose beads were incubated with lysates overnight on a rocking platform at 4 °C. The following morning, the bead–protein complexes were washed, eluted, and examined via Western blot. For proteomic analysis, the cells were treated and then labelled with biotin-PG as described above. To elute the proteins for LC-MS analysis, the bead–protein complexes were washed, eluted in fresh elution buffer (6 M urea, 2 M thiourea, 30 mM biotin, and 2% SDS), and incubated at 42 °C for 1.5 h before centrifugation. Sample lysates were transferred to 10 kDa centrifugal filter units for protein concentration and desalting. The membranes of the filter units were washed with 30 μL of HEPES/arginine buffer and collected in separate tubes. The protein eluates were subjected to Orbitrap LC-MS proteomic analysis at the University of Arkansas Medical Sciences IDeA Proteomics Core.

### 4.4. Filter-Aided Sample Preparation and Orbitrap Eclipse Analysis

Protein samples were reduced, alkylated, and digested using filter-aided sample preparation [52] with sequencing grade modified porcine trypsin (Promega, Madison, WI, USA). Tryptic peptides were then separated by reverse phase XSelect CSH C18 2.5 µm resin (Waters, Milford, MA, USA) on an in-line 150 × 0.075 mm column using an UltiMate 3000 RSLCnano system (ThermoFisher Scientific). Peptides were eluted using a 60 min gradient from 98:2 to 65:35 buffer A:B ratio. (Buffer A = 0.1% formic acid and 0.5% acetonitrile; Buffer B = 0.1% formic acid and 99.9% acetonitrile). Eluted peptides were ionized by electrospray (2.2 kV) followed by mass spectrometric analysis on an Orbitrap Eclipse Tribrid mass spectrometer (ThermoFisher Scientific). MS data were acquired using the FTMS analyzer in profile mode at a resolution of 120,000 over a range of 375 to 1200 *m*/*z*. Following HCD activation, MS/MS data were acquired using the ion trap analyzer in centroid mode and normal mass range with a normalized collision energy of 30%. Proteins were identified by database search using MaxQuant (Max Planck Institute, Munich, Germany) with a parent ion tolerance of 3 ppm and a fragment ion tolerance of 0.5 Da. Scaffold Q+S version 5.3.0 (Proteome Software, Portland, OR, USA) was used to verify MS/MS-based peptide and protein identifications. Protein identifications were accepted if they could be established with less than 1.0% false discovery and contained at least two identified peptides. Protein probabilities were assigned by the Protein Prophet algorithm [53].

#### Proteomics Data Analysis

MS/MS data files were uploaded into Scaffold Q+S software version 5.3.0 for initial analysis. Each treatment group was normalized to its control, and the protein enrichment values are representative of the fold-change ratio. The normalized data sets were exported to Excel for further analysis. Overall protein enrichment was sorted for the 10 and 30 min GnRHa treatment groups compared to the 0 min treatment group. Once sorted, the alternate ID tags for each protein was searched in Enrichr to identify cell signaling pathways containing citrullinated proteins. The top five pathways for each group were identified for 0 to 10 and 0 to 30 min of GnRHa treatment. Additionally, the cytoskeletal proteins actin and tubulin were sorted and analyzed for changes in citrullination enrichment. The top three proteins from each group are represented in heat maps. Major actin-related and MT-related pathways are represented by the top 13 citrullinated proteins. All final data were analyzed in GraphPad Prism software version 9.1.2 as described below.

### 4.5. Western Blots

6X sample buffer consisting of 0.5 M Tris-HCl (pH 6.8), 60% glycerol, 30 mM DTT, and 6% SDS was added into lysates to yield a final concentration of 1X sample buffer and then boiled at 95 °C for 5 min. The samples were subjected to SDS-PAGE using a 10% gel (acrylamide:bis-acrylamide ratio of 29:1) and subsequently transferred to Immobilin PVDF membranes (EMD Millipore, Billerica, MA, USA). The membranes were blocked in 1X casein (Vector Labs) diluted in Tris buffered saline containing 0.1% Tween-20 (TBS-T) overnight at 4 °C. Primary antibodies were incubated overnight at 4 °C: anti-β-actin (1:2000) and anti-β-tubulin (1:4000). The following morning, the membranes were washed in TBS-T, followed by 2 h incubation at room temperature with 1:15,000 anti-rabbit HRP secondary antibody (Jackson ImmunoResearch Labs, West Grove, PA, USA). All blots were washed for 50 min (5 × 10 min) with TBS-T after secondary antibody incubation and then visualized using a SuperSignal West Pico and Femto chemiluminescence substrate (ThermoFisher Scientific). Quantitative densitometry analysis was conducted with BioRad Image Lab software version 4.0. Experiments were repeated at least three times. Means were separated using Tukey’s honest significant difference (HSD) and * indicates significantly different means *p* < 0.05.

### 4.6. Mouse Pituitary Cultures

FVB mice were maintained on a 14 h light–10 h dark cycle with ad libitum access to food and water. Euthanasia was performed in accordance with the guidelines outlined in the Report of the AVMA on Euthanasia. All work in this study was approved by the University of Wyoming Institutional Animal Care and Use Committee (protocol # 20220503BC00550-01, approval 3 May 2022). For primary culture, pituitaries (*n* = 3 per treatment) from mice of 8–16 weeks of age were explanted and dispersed using collagenase and hyaluronidase at 37 °C for 30 min as previously described [9,34]. Dissociated primary pituitary cultures were suspended in culture medium (DMEM supplemented with 10% FBS, 1% non-essential amino acids, 100 IU/mL penicillin, and 100 μg/mL streptomycin) and were plated in 35 mm MaTek dishes and cultured overnight. The cells were then pre-treated with DMSO or 1 µM BB-ClA for 12 h in phenol red-free media containing 2.5% charcoal-stripped FBS followed by treatment with vehicle or 10 nM GnRHa for 0, 10, and 30 min. Primary cultures were examined via immunofluorescent confocal microscopy as described below.

### 4.7. LβT2 and Primary Gonadotrope Immunofluorescence

LβT2 cells were grown in a 35 mm dish with a glass coverslip coated in diluted Matrigel (1:100) and treated with GnRHa and BB-ClA as described above. The cells were fixed with 4% paraformaldehyde supplemented with 2% sucrose and then permeabilized for 5 min at room temperature. The primary and secondary antibodies used in IF are listed above. All samples were visualized on a Zeiss LSM 710 confocal microscope (Carl Zeiss, Jena, Germany) under a 63X oil objective. To image primary gonadotropes, pituitaries were explanted, rinsed free of blood in PBS, and plated. Following treatment, the cells were fixed as stated above and then stained with anti-β-actin (1:100), anti-β-tubulin (1:100), and anti-LHβ (1:50) antibodies diluted in 1X PBS overnight at 4 °C. The cells were washed 3 times for 5 min with 1X PBS and treated with secondary antibodies for 2 h at room temperature. Finally, the cells were washed, stained with DAPI, coverslip mounted, and imaged with a Zeiss LSM 710 confocal microscope under a 63X oil objective.

### 4.8. EB1-GFP Live Cell Imaging

LβT2 cells were grown overnight in 35 mm glass bottom culture dishes coated in diluted Matrigel (1:100). The following day, the cells were transfected with 2.5 µg of the EB1-GFP vector for 48 h using TransIT-2020 Transfection Reagent. The cells were treated with DMSO or BB-ClA for 12 h, followed by a 6 h serum starve in phenol red-free DMEM before imaging. The cells next received 10 nM GnRHa for 0 and 30 min. Following treatment, the dishes were transferred to a mounted Tokai Hit-STX Stage Top Incubator and maintained in 5% CO_2_ at 37 °C in a humidified environment during imaging. Live cell imaging was performed using a complementary metal oxide semiconductor (CMOS) camera (Flash 4.0, Hamamatsu, Shizuoka, Japan) mounted on an Olympus IX81 microscope stand, equipped with a spinning-disk confocal head (CSU-X1, Yokogawa, Tokyo, Japan). Confocal illumination was provided to the system by a LMM5 laser launch (Spectral Applied Research, Richmond Hill, ON, Canada). Integration of all imaging system components was provided by Biovision Technologies (Exton, PA, USA). Image acquisition was performed using Metamorph software version 7.7 (Molecular Devices, San Jose, CA, USA). Images were captured at 0.5 s intervals for 65–94 s using a 60X objective (1.35 NA) immersed in Olympus immersion oil (IMMOIL-F30CC).

#### 4.8.1. Measurement of EB1-GFP Flux

EB1-GFP flux, represented as comets per second (comets^x^s^−1^), was measured by utilizing tools in Fiji software version 2.14.0/1.54f. Each cell (*n* = 10) contained three 10 μm sections (*n* = 30) that were randomly placed using the *line* function. Analysis was performed by two separate individuals, each responsible for the placement of three random sections throughout each cell. With each section, the *reslice* function was then used to create image stacks of the slice over time to create a kymograph. The background fluorescence of each kymograph was averaged and subtracted to increase the accuracy of comet counts. The pixel intensity was then further modified for particle measurement using the *threshold* function to separate pixels in the foreground and background to produce a binary threshold image. The particles representing comets were measured by the *analyze particles* function with the size (pixel^2^) set to 0.15–5, circularity set to 0.00–1.00, and data display set to show particle outlines. The data produced from this included count, total area, average size, % area, mean, and integrated density. Next, a measure of the total binary threshold image was taken, producing the same collection of data as mentioned above. The integrated density from the total binary threshold image was then divided by the integrated density from the particle analysis to provide the number of comets passing through each section. The data were normalized for time by multiplying the comet values by 0.539 s to provide EB1 flux as comets*s^−1^, adjusting for frame rate capture.

#### 4.8.2. Measurement of MT Polymerization Speeds and Lifetimes

MT lengths and lifetimes were measured using a Fiji software version 2.14.0/1.54f kymograph macro plug-in written by J. Rietdorf and A. Seitz as previously used [37,38,39]. EB1-GFP comets labelling growing MT ends, which typically grow with relatively straight tracks, were manually selected for kymograph tracking. The EB1-GFP MT tracks were selected using the *line* function and tracked from the frame before appearance, through to disappearance from the imaging plane. Once selected, the *kymograph* function of the Rietdorf macro plug-in was used to plot the fluorescence over multiple frames to produce the kymograph for the individual comet to represent distance over time. The slope of the EB1-GFP MT track was then analyzed for lengths and lifetimes by utilizing the *read velocities from tsp* function of the macro plug-in to obtain *dx* and *dy* sum values. The *dx* and *dy* sum values were then analyzed in excel and adjusted for the correct dimensions based on pixel size and imaging time. The dx sum values were multiplied by 0.108 (*dx* * 0.108) to convert the values from pixels to μm, while the *dy* sum values were multiplied by 0.539 (*dy* * 0.539) to convert the values from the number of frames to seconds. The converted x-values were used to measure the MT track length and the converted y-values were used to measure the MT track lifetime. EB1-GFP comets were analyzed by two separate individuals (*n* = 15) for a total of *n* = 30 comets.

### 4.9. Statistics

All statistical analyses were performed using GraphPad Prism software version 9.1.2. Data are expressed as means ± SEM of at least three independent experiments. The results were analyzed for significance using one-way ANOVA. Post hoc group comparisons were performed using Tukey’s HSD with the critical value * *p* < 0.05 or ** *p* < 0.01 and *** *p* < 0.001 for declaring significance.

## Figures and Tables

**Figure 1 ijms-25-03181-f001:**
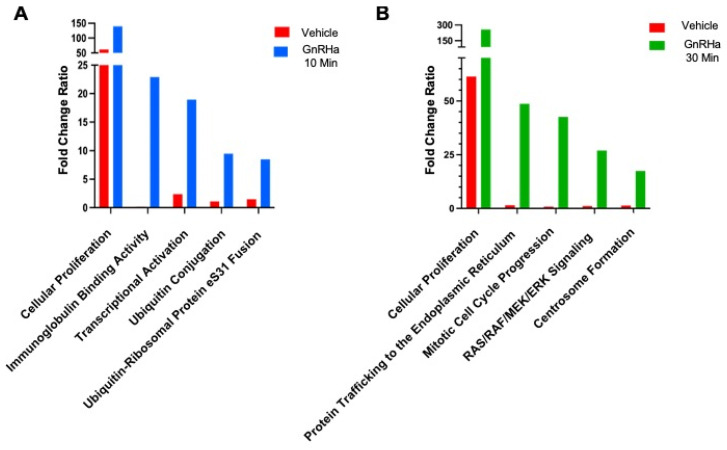
GnRHa stimulation of LβT2 cells induces a temporal citrullinome. LβT2 cells were serum starved and then stimulated with 10 nM GnRHa for 0, 10, or 30 min. Following treatment, cellular lysates were labelled with either vehicle or biotin-PG to enrich the citrullinated proteins. Mass spectrometry was performed and Scaffold Q+S software version 5.3.0 was used to determine the top cellular pathways containing citrullinated proteins following 10 min (blue) (**A**) and 30 min (**B**) (green) of GnRHa treatment compared to baseline citrullination of proteins at 0 min (red).

**Figure 2 ijms-25-03181-f002:**
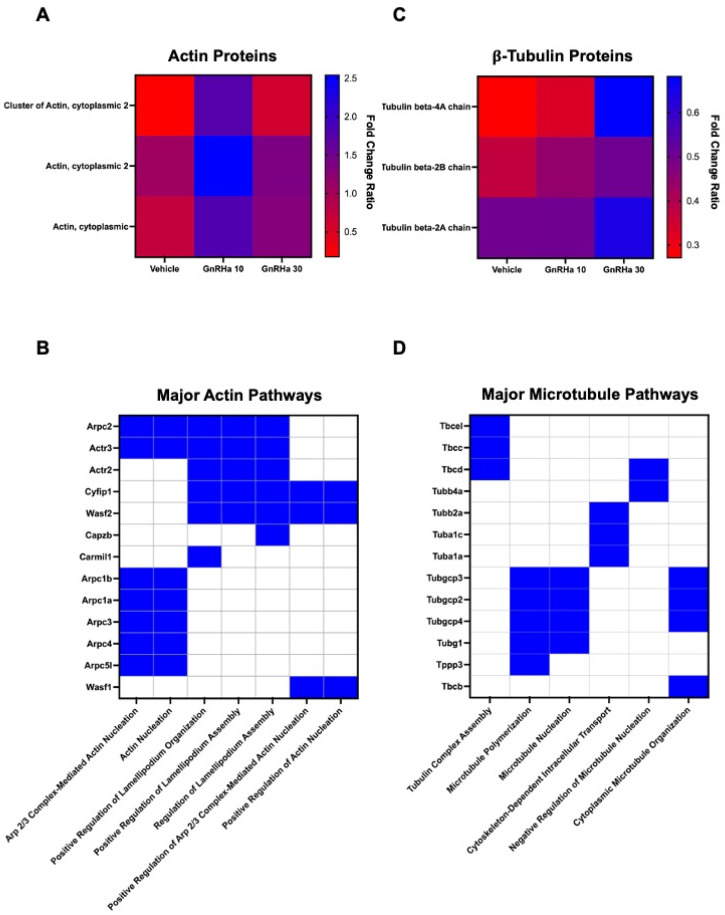
Actin and tubulin are temporally citrullinated following GnRHa stimulation of LβT2 cells. LβT2 cells were stimulated with 10 nM GnRHa followed by enrichment of citrullinated proteins, which were identified via mass spectrometry. Scaffold Q+S software version 5.3.0 was used to determine the changes in the citrullination of actin (**A**) and tubulin (**C**) following 10 and 30 min of GnRHa treatment compared to baseline citrullination of proteins at 0 min. Additional proteins involved in actin (**B**) and MT (**D**) function are also citrullinated. Proteins present in the listed cytoskeletal pathways are designated in blue.

**Figure 3 ijms-25-03181-f003:**
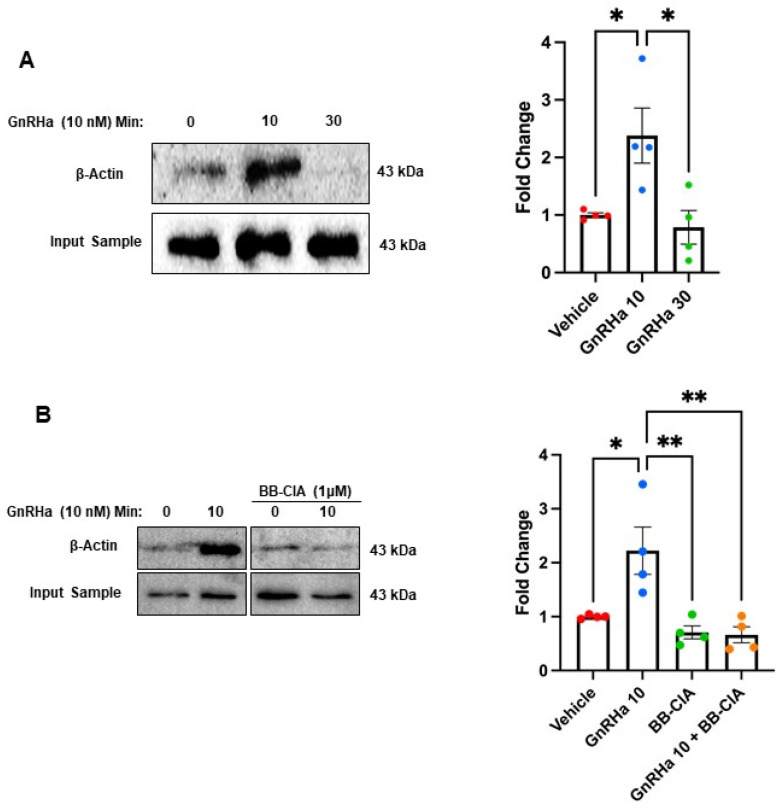
PAD-dependent citrullination of β-actin is elevated following 10 min of GnRHa stimulation. (**A**) LβT2 cells were stimulated with 10 nM GnRHa followed by enrichment of citrullinated proteins. After enrichment, the samples were examined via Western blot using an anti-β-actin antibody. To confirm equal protein loading, 5% of the lysate input samples was also probed with the anti-β-actin antibody. A representative Western blot and quantification from multiple blots (*n* = 4) are shown. The fold change represents the quantitative analysis of the change in β-actin protein saturation normalized to the input sample across treatments as measured using BioRad ImageLab software version 4.0. Means were separated using Tukey’s HSD, and asterisks (*) indicate significant differences (*p* < 0.05). (**B**) LβT2 cells were treated with DMSO or 1 μM BB-ClA for 12 h, and then, samples were generated and examined as described above. A representative Western blot and quantification from multiple blots (*n* = 4) are shown. Quantitative analysis of the blots was conducted and described above (* *p* < 0.05, ** *p* < 0.01).

**Figure 4 ijms-25-03181-f004:**
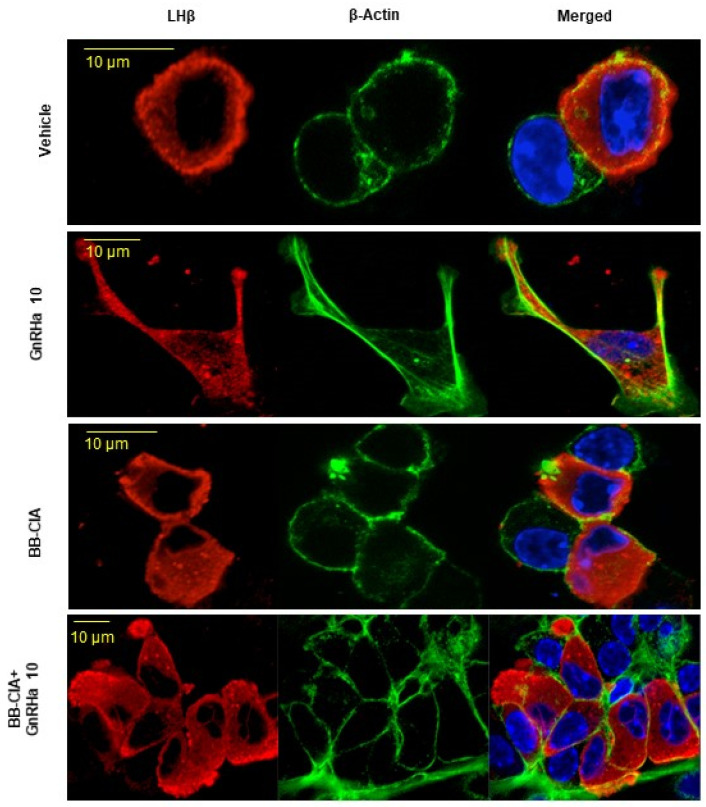
GnRHa-induced β-actin reorganization in primary murine gonadotropes is attenuated following PAD inhibition. Female murine pituitaries were explanted, disassociated, and cultured overnight. Cells were pre-treated with DMSO or 1 µM BB-ClA for 12 h, stimulated with 10 nM GnRHa for 0 and 10 min, then fixed in 4% paraformaldehyde (PFA). Primary cultures were examined via immunofluorescent confocal microscopy using a primary anti-β-actin antibody with a secondary antibody Alexa Fluor 488 (green) and a primary antibody against LHβ with a secondary antibody Alexa Fluor 594 (red) to identify gonadotropes. Nuclei were stained with DAPI. Cells were imaged using a Zeiss LSM 710 confocal microscope under a 40X objective.

**Figure 5 ijms-25-03181-f005:**
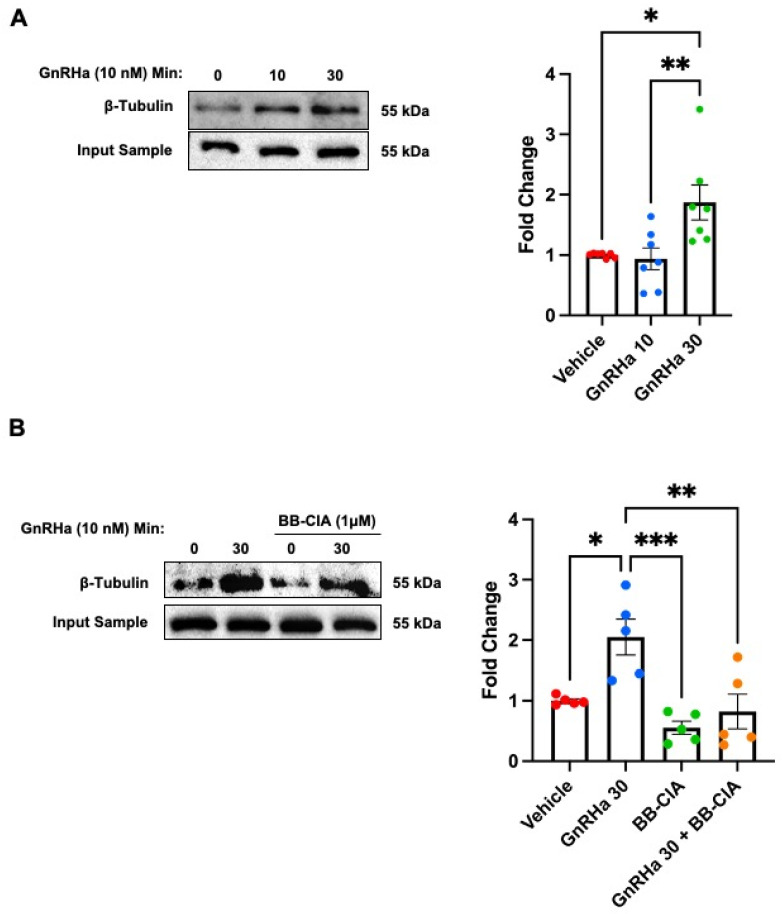
The PAD-dependent citrullination of β-tubulin is elevated following 30 min of GnRHa stimulation. (**A**) LβT2 cells were serum starved and then stimulated with 10 nM GnRHa followed by the enrichment of citrullinated proteins. After enrichment, the samples were examined via Western blot using an anti-β-tubulin antibody. To confirm equal protein loading, 5% input samples were probed with the anti-β-tubulin antibody. A representative Western blot and the quantification of multiple blots (*n* = 7) are shown. Fold change represents the quantitative analysis of the change in β-tubulin protein saturation normalized to the input sample across treatments as measured using BioRad ImageLab software version 4.0. Means were separated using Tukey’s HSD and stars (*) indicate significant differences (* *p* < 0.05; ** *p* < 0.01). (**B**) LβT2 cells were treated with DMSO or 1 μM BB-ClA for 12 h and then, samples were generated and examined as described above. A representative Western blot and the quantification of multiple blots (*n* = 5) are shown. Quantitative analysis of the blots was conducted as described above (* *p* < 0.05; ** *p* < 0.01; *** *p* < 0.001).

**Figure 6 ijms-25-03181-f006:**
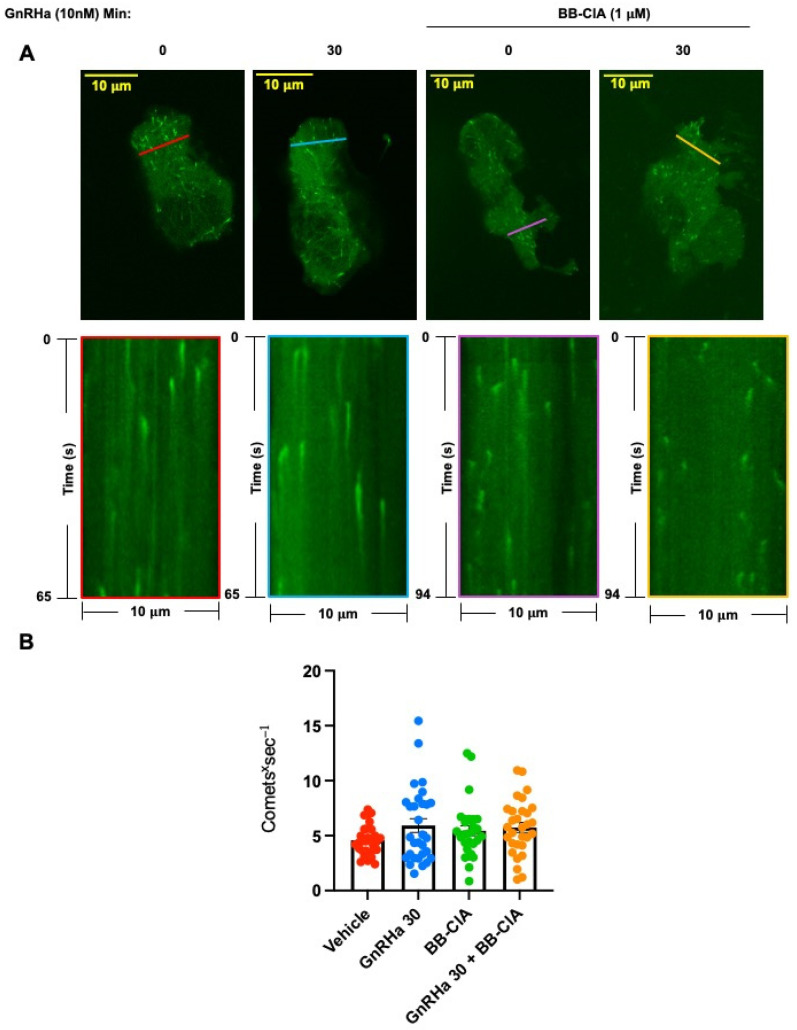
Citrullination does not alter the number of MT comets. (**A**) LβT2 cells were transfected with the EB1-GFP plasmid for 48 hr, pre-treated with DMSO or 1 μm BB-ClA for 12 h, serum starved, and then stimulated with 10 nM GnRHa for 0 and 30 min. The cells were then placed in a Tokai Hit-STX Stage Top Incubator at 37 °C with 5% CO_2_, and images were captured at 0.5 s intervals for 65–94 s by an SDCM using a 60X oil objective. Image stacks were used to analyze the number of EB1 comets passing through 10 μm long sections (*n* = 3 sections) in each cell (*n* = 10 cells; *n* = 30 sections total); see the representative line overlays in the images. Line colors indicate different treatment groups (Red = Vehicle, Blue = GnRHa 30, Purple = BB-ClA, Yellow = GnRHa 30 + BB-ClA). Kymographs (bottom row) show EB1-GFP comets passing transversely through the lines depicted in the corresponding images of cells (top row), with time on the vertical axis and distance on the horizontal axis. (**B**) The bar graph depicts quantitative analysis of the fold change in the flux of EB1-GFP comets, in comets per second, through the 10 μm sections. Each circle represents an individual comet (Red = Vehicle, Blue = GnRHa 30, Green = BB-ClA, Orange = GnRHa 30 + BB-ClA). The difference between the means of each group were separated using Tukey’s HSD and error bars represent the SEM.

**Figure 7 ijms-25-03181-f007:**
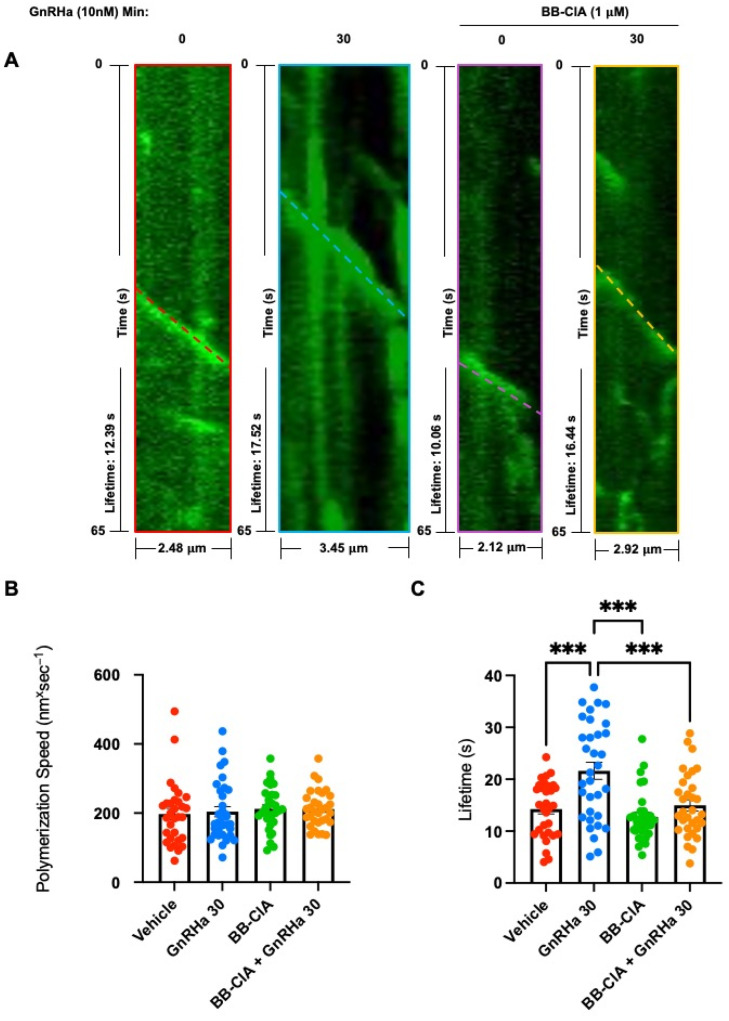
Citrullination does not alter polymerization speed but contributes to GnRHa-induced MT lifetime. (**A**) LβT2 cells were transfected with the EB1-GFP plasmid for 48 hr, pre-treated with DMSO or 1 μm BB-ClA for 12 h, serum starved, and then stimulated with 10 nM GnRHa for 0 and 30 min. The cells were then placed in a Tokai Hit-STX Stage Top Incubator at 37 °C with 5% CO_2_, and live cell images were captured at 0.5 s intervals for 65–94 s by an SDCM using a 60X oil objective. Images are kymographs generated in Fiji software version 2.14.0/1.54f showing MT tracks running parallel (or along) the kymograph’s distance axis. Line colors indicate different treatment groups (Red = Vehicle, Blue = GnRHa 30, Purple = BB-ClA, Yellow = GnRHa 30 + BB-ClA). Bar graphs depict the quantitative analysis of track polymerization speeds (**B**) and track lifetimes (**C**) (*n* = 30) measured in Fiji software version 2.14.0/1.54f using Rietdorff’s kymograph plugin. Each circle represents an individual observation (Red = Vehicle, Blue = GnRHa 30, Green = BB-ClA, Orange = GnRHa 30 + BB-ClA). The difference between the means of each group were separated using Tukey’s HSD. Asterisks indicate a significant difference between groups (*** = *p* < 0.001) and error bars represent the SEM.

## Data Availability

The proteomics data set presented in this study is available in Mass Spectrometry Interactive Virtual Environment (MassIVE) at [doi:10.25345/C5RN30J9J] (accessed publicly on 7 March 2024). The data presented in this study area available from the corresponding author upon request.

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
