# Peer review of "GnRH Induces Citrullination of the Cytoskeleton in Murine Gonadotrope Cells"

_ijms, 2024, doi:10.3390/ijms25063181_

Round 1
Reviewer 1 Report
Comments and Suggestions for Authors
This is a well-written and presented study, investigating the intriguing process of citrullination in the functional response of gonaotrope cells to GnRH stimulation. The authors present data that show that citrullination is involved in reorganization of the cytoskeleton following GnRH challenge, an effect that is blocked with inhibitors of the PAD enzymes. Whilst the majority of the work is performed in the well-characterized LbT2 gonadotrope cell line, the authors also present data generated in primary cultures of mouse gonadotropes that confirms a role for the PAD enzymes in cytoskeletal reorganization in response to GnRH.
The manuscript is sound - I only have a few queries to raise:
1) Is there a specific reason why buserelin was used instead of native GnRH/LHRH? Would you expect a different response to a non-long-acting agonist?
2) In generating samples for MS analyses, how many LbT2 cells were plated, and were multiple samples run and compared for reproducibility? In addition, what passage numbers of LbT2 cells were used for these studies?
3) Figure 4 shows (very nice) images from representative pituitary gonadotropes. How many gonadotropes were imaged for these studies?
4) lines 131-135 Although I understand the need to link previous results to the next data presented, these sentences do seem to be more suited to the discussion.
5) Figure 3 y-axis labels would benefit from a little more detail - 'fold change' of what, exactly?
6) How was the concentration of BB-CIA optimized as part of these studies?
Author Response
Please see attachment. Thank you for your helpful insight.

Reviewer 2 Report
Comments and Suggestions for Authors The main problem studied is the functional consequence of B-tubulin citrullination after GnRH agonist stimulation. This work broaden understanding of the role of PAD catalyzed citrullination in health and disease. This work add new insight on the on the role of GnRH on the citrullination of the cytoskeleton in gonatropes. The methodology is adequate described. The conclusions are adequate to the results described. The references are adequate.
The manuscript is very interesting and methods and results are clearly described. My comments concern the arrangement of the content in three parts ; introduction, results and discussion. In my opinion part Introduction is too long and part of introduction could be transfered to the disscussion. Especialyy these concerning own results of earlier studies. Lines 54-57 could be omited, this is so obvious, that in such article there is no need to deliver so information. In Results discussion of obtained data are very detailed and at least some of it could be moved to the Discussion, which now is too laconic.
Author Response

(The authors gave the same response as above.)

Reviewer 3 Report
Comments and Suggestions for Authors
This outstanding presentation describes the results of GnRH stimulation of L-beta T2 cells with a focus on the Citrillunation of the cytoskeleton. The study is beautifully illustrated, with compelling results that show an important novel role for GnRH in the support of gonadotropin secretion. The only missing link in the study is evidence that LH was actually secreted. Also, from what stage of the cycle were the primary pituitary cells taken? Otherwise it is complete in its portrayal of a role for GnRH in regulating the cytoskeleton.
Author Response

(The authors gave the same response as above.)
